# Analyses of S Protein Homology Using the Genomes of SARS-CoV-2 Specimens Unveil Missing Links in the Temporal Order of Mutations in Its Variants

**DOI:** 10.3390/v15112182

**Published:** 2023-10-30

**Authors:** Ruri Kitayama, Yoshiyuki Ogata

**Affiliations:** Graduate School of Agriculture, Osaka Metropolitan University, Sakai, Osaka 599-8531, Japan

**Keywords:** SARS-CoV-2, Mu variant, temporal order of mutation, prevariant mutation, recombination

## Abstract

(1) Background: Since the emergence of severe acute respiratory syndrome coronavirus 2 (SARS-CoV-2), the evolutionary traits of its variants have been revealed. However, the temporal order of the majority of mutations harbored by variants after the closest ancestors (or precursors), as “missing links”, remains unclear. In this study, we aimed to unveil such missing links based on analyses of S protein homology by focusing on specimens with incomplete sets of S protein mutations in a variant. (2) Methods: Prevariant and postvariant mutations were defined as those before and after the variant’s development, respectively. A total of 6,758,926 and 14,519,521 genomes were obtained from the National Center for Biotechnology Information and the GISAID initiative, respectively, and S protein mutations were detected based on BLASTN analyses. (3) Results: The temporal order of prevariant mutations harbored by 12 variants was deduced. In particular, the D950N mutation in the Mu variant shows V-shaped mutation transitions, in which multiple routes of evolution were combined and resulted in the formation of a V-shaped transition, indicating recombination. (4) Conclusions: Many genome data for SARS-CoV-2 unveiled the candidate precursors of Mu variant based on a data-driven approach to its prevariant mutations in each nation.

## 1. Introduction

Since the emergence of severe acute respiratory syndrome coronavirus 2 (SARS-CoV-2), variants such as Delta and Omicron have been designated by the World Health Organization (WHO). As of April 2023, XBB.1.5, a subvariant (one derived from a variant) of the Omicron variant, is designated by the WHO as the currently circulating variant of interest (VOI); four variants (i.e., Alpha, Beta, Gamma, Delta, and Omicron parent lineage) are classified as previously circulating variants of concern (VOCs), and eight variants (i.e., Epsilon, Zeta, Eta, Theta, Iota, Kappa, Lambda, and Mu) are classified as previous VOIs. The Omicron variant includes various descendant lineages such as BA.1, BA.2, BA.3, BA.4, and BA.5. These variant lineages have been associated in public databases, such as Pango [1,2], GISAID [3], and Nextstrain [4], for construction of phylogenetic trees including the above variants.

Mutations harbored by variants with WHO labels have been reported. For example, the evolution and emergence of N679K and P681H, two Omicron’s S protein mutations, are documented [5,6]. These reports reveal the evolutionary traits of the variants according to whole-genome-based phylogeny. In the phylogenetic tree involving public data or previous reports, however, the temporal order of the majority of mutations occurring in variants after the closest ancestors, as “missing links”, is still unclear. To our knowledge, there are no reports on such order just prior to the development of the virus’s variants.

In this study, we aimed to unveil such missing links about temporal orders of S protein mutations based on analyses of S protein homology using a large number of virus genome specimens and Perl scripts we developed. First, prevariant and postvariant mutations were defined as those before and after a variant’s development, respectively. For this purpose, incomplete sets of S protein mutations harbored by a variant were collected to trace the temporal order of prevariant mutations throughout the world and for each nation.

## 2. Materials and Methods

### 2.1. Data Acquisition

Genome sequences and metadata for SARS-CoV-2 specimens were obtained from the National Center for Biotechnology Information (NCBI) Virus database and the GISAID initiative [3]. The metadata include the accession number, release date, Pango lineage, base length, nucleotide completeness, geological location, and collection date. The number of genomes and specimens was 6,182,920 on 1 September 2022 in NCBI and 14,519,521 on 17 January 2023 in the GISAID initiative. The numbers of specimens by nation are listed in Table 1.

The genomes of all specimens obtained from these databases were used for the present research. Because the metadata of the specimens contained no information on their quality or genome coverage, it was impossible to select the genomes with high quality. Although the genomes obtained through the clinical PCR examinations possibly contain some sequence errors due to their low quality, there are millions of mutation data at each base position, and thus, we determined that influences of such errors were negligible.

A list of mutations in a Pango lineage [1] for each variant with WHO labels (Alpha, Beta, Gamma, Delta, Epsilon, Zeta, Eta, Theta, Iota, Kappa, Lambda, Mu, and Omicron) was obtained by retrieving the variant through cov-lineages.org [7] and outbreak.info [8].

### 2.2. Data Processing

Base sequences of SARS-CoV-2 genomes were used for homology analyses with the BLASTN program from NCBI according to Ogata and Kitayama [9] to detect S protein mutations in each specimen using the Perl scripts we developed. The default parameters of BLASTN were applied. For detection, the genomes mentioned in the previous section were compared with that of the virus’s reference genomes (the accession number in the RefSeq database of NCBI is NC_045512). The pipeline of this processing is described in Ogata and Kitayama [9] and the portal of the Vcorn SARS-CoV-2 database.

### 2.3. Correlation Network between Variants

To visualize variant pairs with common mutations, a correlation network between variants, in which nodes represent variants, was constructed based on the number of mutations shared by each pair of variants. All variants contain D614G, and thus, other mutations were used for constructing the network. Mutations shared by variant pairs were detected using our developed Perl scripts. These Perl scripts are downloadable at the Vcorn SARS-CoV-2 database, i.e., in the website, (1) click the “Overview” tab, (2) click the hyperlink in the last sentence in the Pipeline section, and then click the links in the Folder preparation section in the Pipeline. A correlation network was drawn using the Pajek tool [10]. Pairs of variants sharing two or more mutations were connected. When a variant shares only one mutation with other variants, except for D614G, the variant is located close to the variant with the mutation, even though the variant has no connection in the network.

### 2.4. Correlation Network between Mutations

A correlation network between major mutations, which are contained in 1% or more of all specimens studied, was constructed using recall indices based on specimens that carry each pair of mutations. The recall index of mutation A to mutation B (equivalent to the precision index of mutation B to mutation A) was calculated as a ratio of the number of specimens with both mutations A and B to the number of specimens with only mutation A. If the index is one, all specimens with mutation A contain mutation B. In the correlation network between the major mutations, pairs of mutations with a recall index of 0.9 or more are connected. Considering the index based on a hypothesis that there is no recombination in the virus, mutation A might precede mutation B in the timing of mutations. The detailed procedure is described in our previous reports [9].

### 2.5. Line Charts of Prevariant Mutation

“Prevariant” mutations represent those occurring before evolution of a variant. To trace a history of prevariant mutations occurring in a variant, the ratio of specimens that contain a particular mutation was calculated according to the following steps:First, specimens carrying a complete set of mutations in a particular variant (e.g., in the Alpha variant, 9 mutations) are selected. In these specimens, the ratio of any mutations was one (100%).Specimens that contain all mutations except any single mutation are selected (e.g., in Alpha, 8 mutations). The ratio of specimens that contain a particular mutation to all the selected specimens is calculated for each mutation. If a mutation is the last one to be harbored before a variant’s development, the ratio of the mutation is zero (0%).The previous step is repeated until the set of mutations is not discriminable from those in other variants. For example, 4 mutations (i.e., A570D, T716I, S982A, and D1118H) are carried only by Alpha, and, in contrast, the other 5 of Alpha’s mutations are also harbored in the other variants. Therefore, 6 or more mutations are valid for the calculation for Alpha.

Line charts for Zeta and Theta were unavailable because Zeta mutations are not discriminable from other variants, and there were insufficient specimens for Theta (fewer than 100 specimens) for this analysis. In the case of Omicron, the eight mutations harbored by any of its subvariants (i.e., BA.1, BA.2, BA.3, BA.4, and BA.5) were used to depict its line charts. Grubbs’s test was performed to determine outlier dots in a column that represents specimens with particular numbers of mutations.

### 2.6. Release of Networks and Line Charts from the Vcorn Database

The correlation networks and line charts are published in the Vcorn SARS-CoV-2 database [9]; they were updated for the present mutation analysis every month and for information on COVID-19 every week.

## 3. Results

### 3.1. Relationships between Variants

In the correlation network between variants based on the numbers of mutations shared by variant pairs (at least two mutations) (Figure 1), two variant groups in which variants are tightly connected—i.e., one contains Alpha, Beta, Gamma, Eta, Theta, Mu, and Omicron, and another contains Delta, Epsilon, and Kappa—were obtained. In each group, variants share multiple mutations with each other, possibly due to close relationships and/or recombination. The network shows that Delta’s mutations are quite different from those of Alpha and Omicron. In the correlation network between Omicron’s subvariants, there are two groups of BA.2′s subvariants and a group of BA.4 and BA.5′s subvariants. In contrast, the original Omicron parent lineage, BA.1, and BA.3 appear apart from a network module containing subvariants of BA.2, BA.4, and BA.5. XBF, the subvariant of BA.2, and BQ.1, the subvariant of BA5, share 30 mutations, indicating that XBF is a recombinant between subvariants of BA.2 and BA.5. In fact, XBF is designated a recombinant between BA.5.2.3 and BA.2.75.3 (CJ.1) by the WHO.

### 3.2. Relationships Associated with Convergent Evolution between Mutations

In a correlation network between mutations (Figure 2), there are network modules in which nodes (mutations) are tightly connected to each other and less connected to other nodes, i.e., nodes of network modules representing mutations in Alpha (purple), Gamma (green), Delta (orange), and Omicron and its subvariants (pink, lavender, magenta, red, and maroon). A module in the network indicates that mutations in a variant are well shared with each other in association with convergent evolution and less present in other variants. On the other hand, some mutations are connected to multiple network modules (e.g., P681H, which is present in both modules of Alpha and Omicron’s subvariants), indicating that these mutations occurred in one variant and then were acquired by another variant.

### 3.3. Prevariant Mutations

Figure 3 shows a set of schematic line charts for prevariant mutations, which occurred just before development of the variant. In these charts, the horizontal axes represent the number of mutations in a variant. Namely, a column of a complete set of mutations represents specimens that contain all mutations in the variant, and thus, a dot in the column representing the ratio for a mutation is 100%. Columns −1, −2, and −3 represent specimens that contain all mutations except one, two, and three mutations, respectively; the dots are ratios, as calculated in a manner similar to that in the rightmost column for the complete set. In Figure 3A, the dots are 100% in all columns, indicating that a mutation of interest was present throughout the development of that variant, such as D614G. Figure 3B illustrates dots with very low values in the columns of −1, −2, and −3, and only the dot for the complete set having a value equivalent to 100%, indicating that a mutation of interest was gained as the last mutation for the variant. Similarly, Figure 3C indicates that a mutation of interest was added as the second-to-last mutation for the variant. In contrast, Figure 3D depicts that the dots gradually increase. In this case, a mutation of interest may be unstable or difficult to detect through homology search by the BLAST program.

Figure 4 shows line charts of prevariant mutations in 11 variants with WHO labels. These line charts reveal the temporal order of certain prevariant mutations for variants as shown in Table 2 and in the following subsections.

#### 3.3.1. Alpha

Nine mutations are harbored in the Alpha reference genome, and 489,389 specimens carry these mutations. Five of the mutations are also present in the reference genomes of the other variants; there are four mutations that are original to Alpha. In the case of Alpha (Figure 4A), specimens sharing I68fs (i.e., frameshift at the 68th isoleucine) and V143fs (i.e., frameshift at the 143rd valine) increase from approximately 10% to 60%. Based on these increases, it was deduced that I68fs and V143fs were the last and second-to-last prevariant mutations, as based on Grubbs’s test for the ratios in Columns 7 and 8 (α = 0.001). Before these mutations, it is difficult to speculate the order of mutations, e.g., in Column 6, the specimen shares of any dots are below 100% (approximately 60 to 80%) because the specimens related to the column carry only a mutation original to Alpha, and thus, they can be classified as other variants.

#### 3.3.2. Beta

In the reference genome of Beta, eight mutations are present, and 5355 specimens carry these mutations. Five mutations are original to Beta. Grubbs’s test for Beta (*α* = 0.001) showed T240fs (i.e., frameshift at the 240th threonine) to be present in a low number of specimens with five, six, and seven mutations (Table 2). D215G was present in a low number of specimens with six mutations as well as T240fs, resulting in a kind of V-shaped mutation transition between five and seven (Figure 4B). The ratios of E484K and N501Y were lower than those of other mutations in specimens with five mutations. Dissimilar to Alpha, it is difficult to deduce the order of prevariant mutations, due to several V-shaped mutation transitions and the smaller number of specimens than Alpha (e.g., less than 100 specimens in Columns 4 and 5).

#### 3.3.3. Mu

In the reference genome of Mu, nine mutations are found, and 4428 specimens carry these mutations. Three mutations originate in Mu. The D950N mutation in Mu shows a V-shaped mutation transition between seven and nine in mutation numbers (Figure 4C), i.e., the mutation is carried by almost all specimens with seven and nine mutations but in approximately 60% of specimens with eight mutations. Although Y144S and Y145N are not present in almost all specimens with seven mutations, they are harbored by almost all specimens with nine mutations (i.e., a complete set of Mu’s mutations).

To unveil the development of Mu, prevariant mutations for individual nations in which many specimens were obtained were analyzed using the GISAID dataset, as described in the next section.

#### 3.3.4. Gamma

The Gamma reference genome harbors 12 mutations, and 22,667 specimens carry these mutations. Seven mutations are original to Gamma. There was no statistically significant transition in Gamma’s mutations (Figure 4D), i.e., all mutations show gradual transitions. V1176, E484K, and N501Y seem to show differences between specimens with 10 and 11 mutations, and L18F, T20N, and P26S seem to show differences between specimens with 9 and 10 mutations, but without statistical significance. These nonsignificant transitions are possibly due to the low number of specimens, such as those in Columns 6 to 8 (i.e., fewer than 100).

#### 3.3.5. Delta

The reference genome of Delta shows seven mutations, and 2,271,968 specimens carry these mutations. Two mutations are original to Delta. D950N and E156fs (i.e., frameshift at the 156th glutamic acid) are present in a low number of specimens with six mutations (Figure 4E). Based on these changes, it was deduced that D950N and E156fs were the last and second-to-last prevariant mutations, respectively, based on Grubbs’s test for the ratio for six mutations (α = 0.05). Although there were more Delta specimens than for other variants, except for Omicron, there is less information on the order of prevariant mutations, because the number of its original mutations is insufficient (i.e., only three mutations).

#### 3.3.6. Omicron

As there are five reference genomes of Omicron (i.e., BA1, BA2, BA3, BA4, and BA5), seven mutations that are commonly present in these subvariants were selected for their prevariant transition and are harbored by 1,715,596 specimens (Figure 4F). Four mutations are original to Omicron. Almost all of the specimens with six mutations carry only six mutations except for S373P and T478K, indicating that these mutations were the last and second-to-last prevariant mutations. The number of specimens with six mutations totaled 990,711 (57.7% of specimens with the complete set); the number of specimens with seven mutations was 26,911 (1.5% of those with the complete set). This indicates both mutations to be harbored almost simultaneously. Over 20 mutations are harbored by Omicron’s subvariants mentioned above.

### 3.4. Mu’s Prevariant Mutation by Nation

Figure 5 shows Mu’s prevariant mutations in the whole datasets from NCBI and GISAID and for nations in which there are many (over 100 specimens in either column in a chart of prevariant mutation) specimens with Mu’s mutations. In comparison with the whole datasets from NCBI and GISAID, the shares of Y144S and Y145N in both datasets are very low in specimens with seven mutations, despite small differences such as Y144S, and the share of D950N shows a V-shaped transition. The tendency in the United States was quite similar to that in the NCBI datasets, because the majority of the NCBI specimens were obtained from the United States (47%) and the United Kingdom (33%). In the transition of D950N, V-shaped tendencies were detected in the NCBI dataset (Figure 5A), the GISAID dataset (Figure 5B), the United States (Figure 5C), Spain (Figure 5E), and Ecuador (Figure 5F).

In contrast, Colombia (Figure 5D) showed very high shares through the transition, with low shares in specimens with seven and eight mutations in Mexico (Figure 5G) and Chile (Figure 5H), especially in Chile, in which 823 specimens with eight mutations were studied. In the transition of Y144S and Y145N, the shares of specimens with seven mutations are very low in the NCBI dataset, the GISAID dataset, and in the United States, Colombia, Spain, Ecuador, and Panama (Figure 5I), especially in Colombia (363 specimens). The shares of both mutations in the specimens with eight mutations are approximately half, indicating that either mutation in these specimens might be undetected through the present homology analysis. For example, although D614G is present in all variants, the mutation was not necessarily detected in all specimens studied through the present homology analysis. 

In contrast, those in Mexico and Chile show that the shares of only Y145N are very low in specimens with seven mutations and very high in those with eight mutations. When the chart of Colombia is combined with that of Chile, D950N shows a V-shaped transition, such as those of the NCBI and GISAID datasets.

This indicates that the development of the Mu variant can be based on two ways (Figure 6), i.e., one in Colombia (Figure 5D) and Panama (Figure 5I), and another in Mexico (Figure 5G) and Chile (Figure 5H).

## 4. Discussion

Many genome data for SARS-CoV-2, which are available in public databases, are useful to trace a missing link in mutational transition. In the present study, by preparing specimens with incomplete sets of mutations carried by a variant, the temporal order of prevariant mutations for that variant was deduced. If the number of specimens is very low, it is difficult to determine the order of mutations based on statistical significance, as in the cases of Zeta and Theta. In the present algorithm, the temporal order of mutations is statistically deduced by considering the numbers of sequences. The mechanism of proteolytic cleavage (6) and the mutational profile on the evolution of multiple variants are discussed in a previous report on N679K and P681H, two of Omicron’s mutations. However, the temporal order of prevariant mutations was not the focus of the study.

Information on the origin of the Omicron variant and its subvariants is published in several databases, such as cov-lineages.org [7] and NCBI Virus. In cov-lineage.org, the earliest specimens of BA1, BA2, BA3, BA4, and BA5 are dated 25 June 2020, 28 March 2020, 23 November 2021, 2 July 2020, and 20 July 2020, respectively. Approximately seven S protein mutations are shared by Omicron’s subvariants. According to the Vcorn SARS-CoV-2 database, D614G is present in all variants, H655Y only in Gamma and Omicron, N679K mainly in Omicron (also in some specimens of Alpha, Gamma, and Delta), P681H mainly in Alpha, Theta, Mu, and Omicron, Q954H mainly in Omicron (also in some specimens of Delta), and N969K mainly in Omicron (also in some specimens of Delta). Among these mutations, N679K, Q954H, and N969K are shared by some Delta specimens, indicating the possibility of recombination between two different variants. On the other hand, it is difficult to discuss the disappearance of other variants based on the data in the present study.

There is also a missing link in the temporal order of mutations between SARS-CoV-1 and SARS-CoV-2. There are many previous reports on evolution before SARS-CoV-2 [11]. Although there are insufficient genome data for the variant lineage between them, if such genome data can be obtained, the temporal order of mutations will be deducible. The present approach is applicable to deduce the temporal order of other viruses and organisms, if a large amount of genome data are available.

The transition in Alpha’s mutations reveals that two mutations (I68fs and V143fs) were gained just before the variant’s development, as based on Grubbs’s test for incomplete sets of specimens. V-shaped transitions, such as those found for Mu, were observed in mutation transitions of multiple variants. As described in the Mu’s prevariant mutation by nation section, when the transitions of Colombia (Figure 5D) and Chile (Figure 5H) are combined, D950N shows a V-shaped transition. For a reasonable interpretation of such transitions, we hypothesize the following steps: (1) specimens carry six of a complete set of Mu’s mutations except for D950N, Y144S, and Y145N; (2) D950N was gained by some of these specimens (seven mutations total), whereas Y144S and Y145N were gained by other specimens (eight mutations in total); and (3) Y144S and Y145N were gained by specimens harboring D950N (nine mutations in total), and D950N by those with Y144S and Y145N (nine mutations in total). In the third step, recombination may explain the V-shaped transition. Recombination between multiple Omicron subvariants is reported in cov-lineages.org [7], e.g., XB, XE, and XAM. Based on this hypothesis, a line chart showing a mutation transition will support detection of recombination between virus genomes as well as the temporal order of the transition. However, the hypothesis based on the genomes obtained from various nations does not indicate that the possibility of linkage disequilibrium and convergent evolution can be ignored.

An approach to trace prevariant mutations by nation is useful to obtain statistical evidence for recombination in the Mu variant. The transitions of Mu’s prevariant mutations in Colombia and Chile are characteristic of such evidence to unveil Mu’s precursors. The transition in Colombia (Figure 5D) shows very low share of Y144S and Y145N in specimens with seven mutations and shows high shares of D950N throughout the transition. The transition in Chile (Figure 5H) shows an obvious increase in the share of Y145N between specimens with seven and eight mutations and very low D950N shares in specimens with seven and eight mutations. These transitions indicate that the genomes in specimens with seven mutations in Colombia and the genomes in specimens with eight mutations in Chile may be candidate precursors of the Mu variant (Figure 6). Although another mutation might be present in each precursor based on convergent evolution, if particular hosts are infected by these candidates at the same time, the Mu variant might have developed in hosts through recombination. Although there are a few reports in which Y144S, Y145N, and D950N were studied [12,13], their functionality and neutrality were not discussed. Therefore, the significance of the precursors with these mutations is still unclear. Oliveira et al. [14] indicated that international travel after a major international soccer event in Brazil in July 2021 might have contributed to the introduction and spread of this variant. However, the number of virus specimens is limited in comparison with that of positive cases published by the WHO. In Colombia, for example, the accumulated number of positive cases is over 6 million; in contrast, that of genomes obtained in the GISAID Initiative was 23,235 at January 2023, i.e., approximately 0.3%. More specimens of virus genomes will lead to more reasonable statistical evidence to reveal the missing link in the temporal order of mutations.

## 5. Conclusions

By preparing specimens with incomplete sets of mutations carried by a SARS-CoV-2 variant, the temporal order of prevariant mutations for that variant was statistically deduced. The deduced order for the Mu variant suggests that there were two kinds of the variant precursors in different nations of South America, and thus, it indicates that the variant developed due to recombination, though the possibility of linkage disequilibrium and convergent evolution are not ignored. Such temporal order can be deduced by an approach using a phylogenetic tree based on multiple sequence alignment. However, the order depends upon algorithms for the approach, and in general, the tree is unsuitable for depicting evolution based on recombination due to its bifurcated structure. A large amount of data of the genomes of a particular virus can lead to statistically deducing the temporal order of its mutations and the recombination in its development.

## Figures and Tables

**Figure 1 viruses-15-02182-f001:**
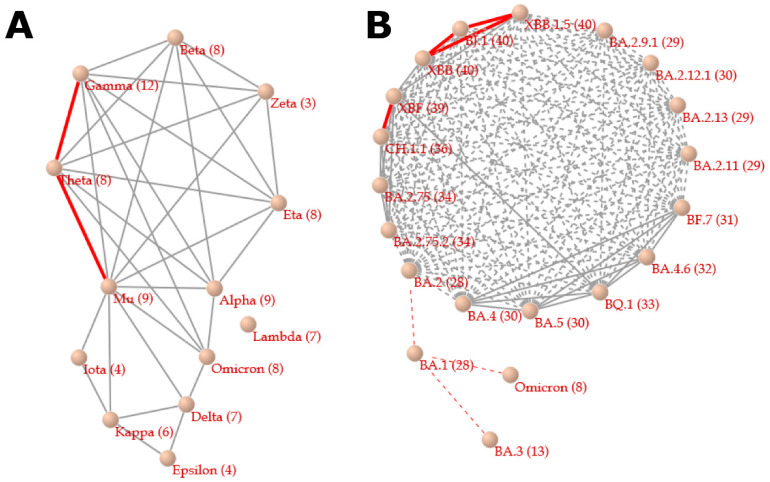
Correlation network between variants (**A**) and between Omicron’s subvariants (**B**). In the network between variants, nodes represent variants with WHO labels and are connected to other variants when they share 4 or more (red, thick lines) and 2 or 3 (gray, thin lines) mutations. Among seven mutations, Lambda has a single mutation (i.e., D614G) shared with the other variants. In the network between Omicron’s subvariants, nodes represent the subvariants and are connected when they share 35 or more (red, thick, solid lines), 30 to 34 (gray, thin, solid lines), 25 to 29 (gray, thin, dashed lines), and 24 or fewer (red, thin, dashed lines) mutations.

**Figure 2 viruses-15-02182-f002:**
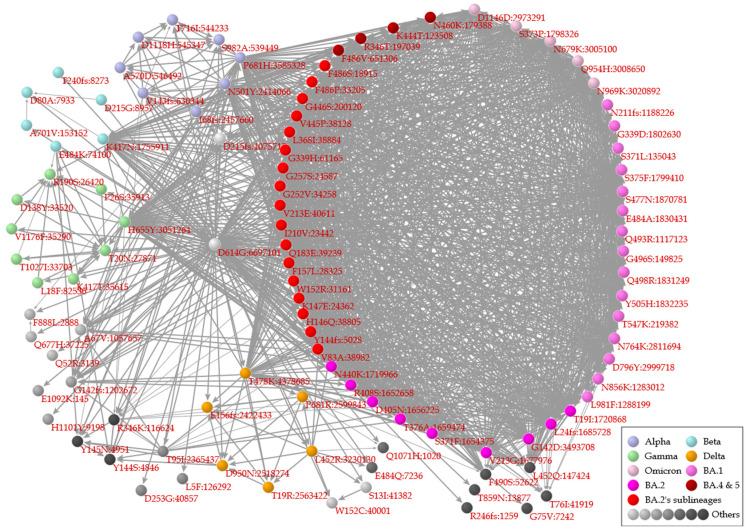
Correlation network between major mutations. Nodes represent mutations connected to other mutations based on recall indices (0.9 or higher) between pairs of mutations. The color of the nodes indicates Alpha (purple), Beta (blue), Gamma (green), Delta (orange), Omicron (pink), BA.1 (lavender), BA.2 (magenta), BA.2′s sublineages (red), BA.4 and BA.5 (maroon), and others (gray). In a label for a mutation, the number after a colon represents the number of specimens carrying the mutation. Each arrow is depicted from a mutation harbored in larger specimens to a mutation in fewer specimens.

**Figure 3 viruses-15-02182-f003:**
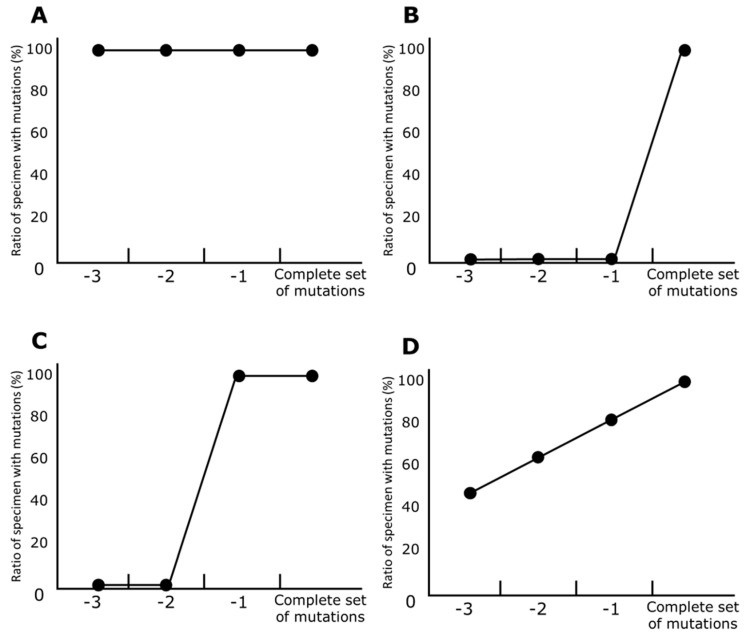
Schematic line charts for prevariant mutations. The horizontal axis represents the numbers of variant mutations contained in specimens. The rightmost column represents specimens with a complete set of a variant’s mutations, and columns −1, −2, and −3 represent all of its mutations except for 1, 2, and 3 mutations, respectively. Dots represent the ratio of specimens with a mutation to all specimens in each column. (**A**) Dots are 100% in all columns, indicating that a mutation of interest was present throughout the development of that variant, such as D614G. (**B**) Dots with very low values in the columns of −1, −2, and −3, and only the dot with the value equivalent to 100%, indicating that a mutation of interest was gained as the last mutation for the variant. (**C**) Dots with very low values in columns −2 and −3 and two dots with values equivalent to 100%, indicating that a mutation of interest was added as the second-to-last mutation for the variant. (**D**) Dots that gradually increase, indicating that a mutation of interest may be unstable or difficult to detect through homology search by the BLAST program.

**Figure 4 viruses-15-02182-f004:**
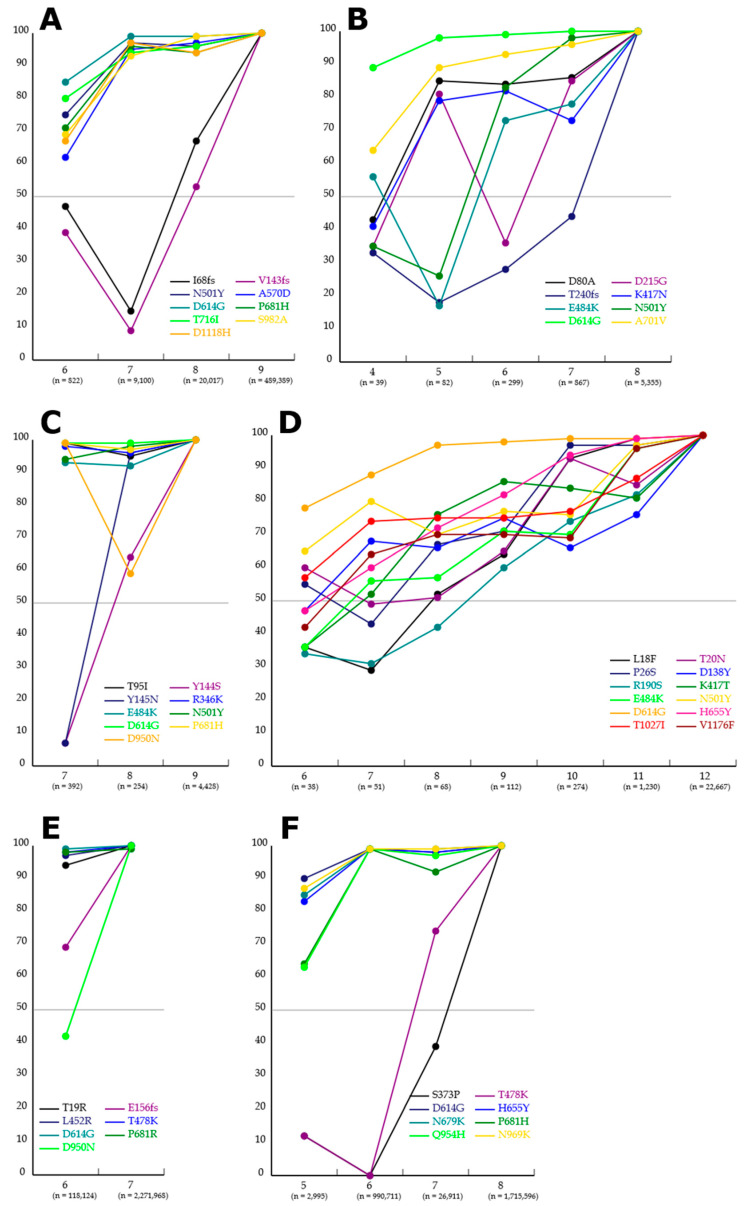
Line charts for prevariant mutations for 6 variants, i.e., Alpha (**A**), Beta (**B**), Mu (**C**), Gamma (**D**), Delta (**E**), and Omicron (**F**), based on the datasets obtained from the NCBI Virus database. The horizontal axis represents the numbers of incomplete sets (except for the rightmost column) and complete set (in the rightmost column) of mutations harbored in a variant. The vertical axis is similar to those in Figure 3. The number of specimens in each column is shown in parentheses in the column.

**Figure 5 viruses-15-02182-f005:**
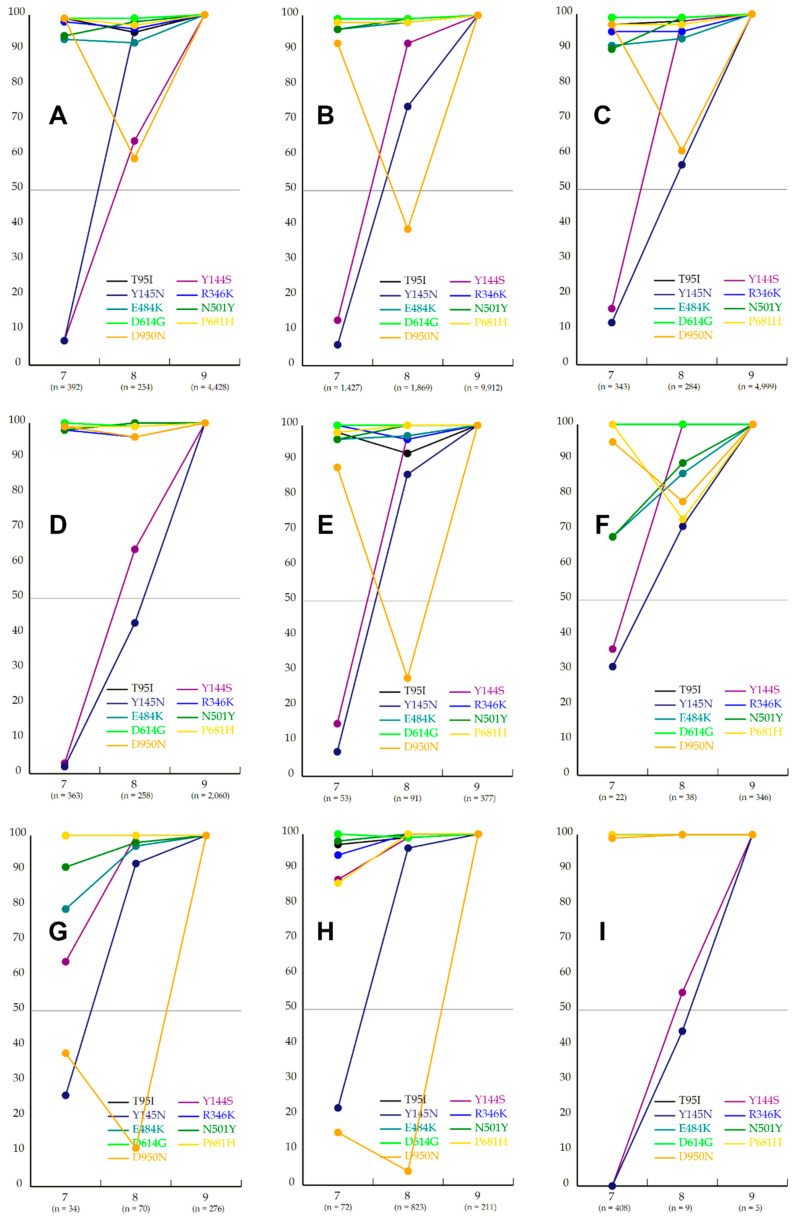
Line charts for Mu’s prevariant mutations in the NCBI dataset (**A**), the GISAID dataset (**B**), and in the United States (**C**), Colombia (**D**), Spain (**E**), Ecuador (**F**), Mexico (**G**), Chile (**H**), and Panama (**I**). Both axes should be interpreted as in Figure 4. The number of specimens in each column is shown in parentheses in the column.

**Figure 6 viruses-15-02182-f006:**
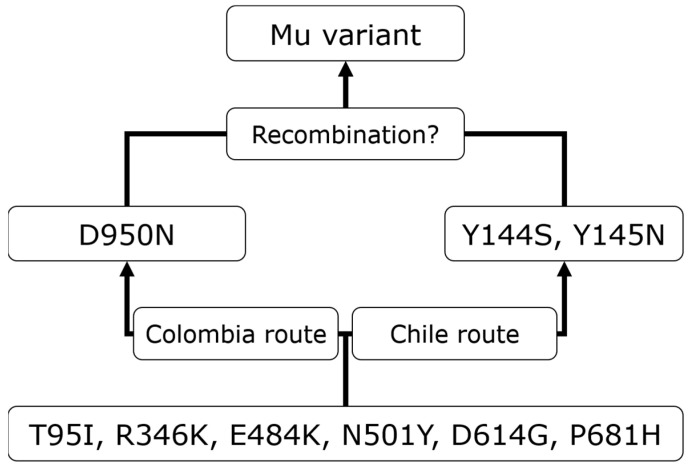
The diagram deduced according to Mu’s prevariant mutations. The Colombia route includes Colombia and Panama, and the Chile route includes Chile and Mexico.

**Table 1 viruses-15-02182-t001:** A list of the numbers of the positive cases reported by the WHO (WHO) and the specimens by nation in the datasets obtained from the NCBI Virus database (NCBI) and the GISAID Initiative database (GISAID).

Nation Name	WHO 2023-04-12	NCBI 2023-03-01	GISAID 2023-01-17
United States	102,873,924	3,275,120	4,463,651
United Kingdom	24,330,379	2,249,734	2,919,535
Germany	38,368,891	512,215	836,315
Denmark	3,409,630	403,025	619,871
France	38,791,479	48,596	563,393
Japan	33,523,927	8557	512,329
Canada	4,634,277	241	492,655
India	44,768,172	4396	261,507
Sweden	2,702,703	203	231,635
Austria	6,046,956	300	220,750
Spain	13,798,747	1190	186,336
Brazil	37,319,254	16,392	185,932
Belgium	4,782,863	32	169,910
Australia	11,153,745	13,356	160,828
Switzerland	4,399,085	152,642	160,353
Italy	25,715,384	1547	157,212
Netherlands	8,610,372	1590	155,330
Israel	4,817,255	134	146,466
South Korea	30,918,060	413	110,902
Turkey	17,004,677	204	101,482
Ireland	1,708,435	19	99,966
Mexico	7,553,646	15,981	81,908
Slovenia	1,342,156	0	81,120
Poland	6,504,340	1219	74,754
Norway	1,481,760	6	74,212
Czech Republic	4,636,282	29	51,963
Luxembourg	319,959	1	50,289
South Africa	4,072,533	5405	46,343
Finland	1,468,123	52	44,775
Slovakia	1,865,828	36,222	44,681
Indonesia	6,752,606	76	44,100
Portugal	5,577,825	33	42,133
Croatia	1,271,276	1	41,765
Lithuania	1,316,086	0	40,477
Chile	5,272,767	818	39,882
Malaysia	5,052,337	434	35,404
Thailand	4,728,967	5467	35,118
Russia	22,727,542	565	30,453
New Zealand	2,217,047	16,636	28,620
Singapore	2,298,689	11	28,126
Philippines	4,083,678	109	25,685
Greece	5,972,760	98	23,839
Colombia	6,363,058	367	23,235
Bulgaria	1,301,475	2	20,389
Romania	3,380,891	109	17,851
Reunion	494,595	0	17,784
Latvia	977,172	0	17,567
Hong Kong	-	2518	16,781
Estonia	617,247	1681	14,913
Puerto Rico	1,110,017	3345	13,382
Iceland	209,191	25,459	12,903
Kenya	342,992	8178	12,334
Argentina	10,044,957	254	11,981
Bahrain	696,614	12,130	10,304
Ecuador	1,059,529	7	8920
Costa Rica	1,226,315	0	8750
Vietnam	11,527,745	2525	8173
Nigeria	266,675	3161	7906
Canary Islands	-	0	7599
Bangladesh	2,038,091	1853	7528
Peru	4,492,891	755	6637
Panama	1,033,781	0	6536
Qatar	502,436	24	6161
Senegal	88,978	0	5961
Pakistan	1,580,021	1524	5806
Mauritius	298,099	0	5776
French Guiana	98,041	0	5469
Botswana	329,837	0	5176
Brunei	284,632	0	4861
Ghana	171,527	297	4636
Papua New Guinea	46,837	0	4492
Trinidad and Tobago	191,007	0	4391
Iran	7,597,982	1709	4266
China	99,239,252	695	4209
Cambodia	138,726	2	3854
Martinique	229,479	0	3795
Guatemala	1,244,812	10	3733
Aruba	44,114	0	3613
Sri Lanka	672,092	6	3564
Nepal	1,001,951	19	3523
Egypt	515,913	1191	3202
Gibraltar	20,550	0	3029
Paraguay	735,759	298	2896
Cyprus	655,664	0	2690
Sint Maarten	11,030	0	2673
United Arab Emirates	1,058,979	0	2630
Georgia	1,836,791	21	2607
Taiwan	10,239,690	380	2602
Lebanon	1,235,177	931	2575
Jamaica	154,602	438	2446
Guadeloupe	202,163	0	2437
Curacao	45,798	0	2128
Democratic Republic of the Congo	95,944	0	2091
Liechtenstein	21,460	1804	1952
Bonaire	11,885	0	1913
Zambia	343,415	1	1891
Namibia	171,222	1	1886
Dominican Republic	660,937	386	1762
Morocco	1,272,733	22	1750
Tunisia	1,152,033	278	1732
Kosovo	273,764	0	1710
Cuba	1,112,853	2	1600
Seychelles	50,937	1031	1551
Jordan	1,746,997	28	1549
Kazakhstan	1,501,450	360	1514
Bosnia and Herzegovina	402,636	0	1513
Gambia	12,622	467	1493
Uganda	170,515	116	1428
Malawi	88,620	289	1379
Angola	105,353	0	1325
Cameroon	124,834	131	1322
Maldives	185,894	0	1292
Mayotte	42,008	0	1258
Iraq	2,465,545	669	1222
Ukraine	5,484,936	0	1202
Kuwait	665,527	15	1182
Benin	28,014	265	1176
Saudi Arabia	836,442	1092	1168
Suriname	82,495	1	1124
Mozambique	233,334	0	1098
Serbia	2,524,670	153	1093
Eswatini	74,520	0	1051
Belize	70,782	4	1030
Venezuela	552,398	40	1024

The list contains nations with 1000 or more GISAID specimens.

**Table 2 viruses-15-02182-t002:** List of the temporal order of prevariant mutations based on Grubbs’s test.

Variant	Mutation Number	Statistical Significance *	Mutation List
Alpha	8	***	V143fs, I68fs
Alpha	7	***	V143fs, I68fs
Beta	7	*	T240fs
Beta	6	*	T240fs, D215G
Beta	5	**	E484K, T240fs, N501Y
Delta	6	*	D950N, E156fs
Epsilon	3	*	W152C
Eta	7	*	Q52R
Eta	6	*	V143fs, Q52R
Mu	8	***	D950N, Y144S
Mu	7	***	Y145N, Y144S
Omicron	7	***	S373P, T478K
Omicron	6	**	S373P, T478K
Omicron	5	*	S373P, T478K

The numbers of asterisks represent Grubbs’s test significance levels (i.e., *** 0.001, ** 0.01, and * 0.05).

## Data Availability

The datasets used for the present research are available from WHO Coronavirus Dashboard (https://covid19.who.int/WHO-COVID-19-global-data.csv) weekly, SARS-CoV-2 Data Hub in NCBI Virus database (https://www.ncbi.nlm.nih.gov/labs/virus/vssi/#/sars-cov-2) monthly, and GISAID (https://gisaid.org/) bimonthly. In the website of GISAID, user registration (for free) is required for downloading a bulk of genome datasets. The latest version of the Vcorn SARS-CoV-2 database was released at 31 August 2023.

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
