# Peer review of "Analyses of S Protein Homology Using the Genomes of SARS-CoV-2 Specimens Unveil Missing Links in the Temporal Order of Mutations in Its Variants"

_viruses, 2023, doi:10.3390/v15112182_

Round 1

Reviewer 1 Report

Comments and Suggestions for Authors

This is a nice paper by Kitayama and Ogata who analyzed specimens with incomplete sets of mutations in a SARS-CoV-2 variant to statistically deduce the temporal order of pre-variant mutations for that variant S protein. For example, the deduced order for the Mu variant suggests that there were two kinds of precursors in different nations of South America, suggesting that the variant developed due to recombination. This type of computational and statistical analysis, although quite complex, reveals interesting aspects of the COVID-19 virus evolution. In sum, big genomic data for SARS-CoV-2 revealed the candidate precursors of Mu variant based on a data-driven approach to its pre-variant mutations in various nations. The analysis is well performed and appears complete. Figures and legends are clear and convincing.

COMMENT:

Would it be possible to prepare and add a schematic representation of the evolution of SARS-CoV-2 based on the data/conclusions of this paper? A diagram that would help the lay person to understand the scope of the study. That would be nice, but maybe it is not feasible. I let the authors decide what can be done.

Reviewer 2 Report

Comments and Suggestions for Authors

The authors report an analysis of mutation dynamics leading to the emergence of S variants using network analysis. While the approach is interesting, in my opinion, the work is affected by a few limitations.

The authors apply network analysis without apparently using quantitative parameters to describe network properties and the properties of single or group of nodes.

The conclusions should indicate also how the results obtained can be useful to the study of the virus and pandemic evolution

The number of references is too low to give a fair overview of the pertinent work published.

Quality of Figures should be improved.

Comments on the Quality of English Language

I suggest the the authors revise extensively the english syntax as several sentences are difficult to understand. For example, the definition of recall index (lines 95-99) is not clear, in my opinion.

Reviewer 3 Report

Comments and Suggestions for Authors

Ruri Kitayama and Yoshiyuki Ogata presented very interesting work in their article. The work is devoted to the theoretical analysis of the evolution of variants of SARS-CoV-2. This is a very important topic, since sometimes the nature of the occurrence of a particular variant remains not entirely clear. Sometimes it is not possible to detect missing transitional variants of the virus. The authors chose the sequence encoding the spike protein as a target for analysis. This is understandable, because it is this protein that is the target of humoral immunity. I really liked that the authors tried to trace co-evolution.

The authors have done a great job. All illustrations and tables are designed correctly. The article contains a discussion section, where the authors reveal the outline of the study.

The conclusions are correct and based on the results of the study.

However, I still have some minor comments:

The introduction to the article is very concise. Still, I would like to understand the importance of this work in the context of existing research and the current situation.

Reviewer 4 Report

Comments and Suggestions for Authors

The authors present an interesting analysis of over 20 million genomes to map the temporal order of mutations in the S protein for 12 COVID variants. This is relevant from several perspectives. It may help us understand better the evolution of the virus, the dynamics of the specific mutations that led to the emergence of different strains, and how the order of the mutations may have affected the function of the S protein as well as the infectivity of the variants.

However, in its current version, the paper lacks clarity and raised the following issues that, in my opinion, need to be clarified before the paper is suitale for publication.

-----

Abstract: (minor) I see no point in mentioning the format of a sequence file in the abstract. Giving the number of genomes analyzed is enough.

Page 1-2. Introduction. The authors argue that the temporal order of the mutations is still unclear, but there are no references or mention of previous attempts to tackle this problem. If this is the first attempt (to the knowledge of the authors), the authors should say it explicitly. Otherwise, this is the right place to explain why previous attempts have been unsuccessful.

Page 5. Methods section “Data Processing”. This section is poorly explained. This is where the authors are supposed to describe their algorithmic approach and provide technical details. It says that the pipeline is described in the Vcorn SARS-CoV-2 database but there is no reference or URL here. I went to the website, under the Overview tab, and the description of the pipeline is six steps with one sentence summary descriptions each. Relevant steps in the pipeline (i.e. 2, 4, 5, and 6) lack technical details or links to the corresponding explanations. If the technical description of the pipeline is in the 2022 paper by Yoshiyuki Ogata and Ruri Kitayama. The reference should be given in this section. I mean, what is the minimum sequencing coverage of the genomes used (e.g., 50X, 100X)? What decisions had to be made when preprocessing the data (quality control)? What BlastN parameters were used? Etc. etc.

Page 5. Lines 84-85. Methods section “Correlation network between variants”. Just indicating that the mutations were “detected using our developed Perl scripts” is not useful unless the authors point the reader to the proper description or provide references. It is evident hat the authors have done a lot of work, but it would be impossible for others to reproduce this analysis with such a poor level of description. If the technical descriptions are lengthy, the authors can add it as supplementary material.

Page 5. Page 5, line 111. Methods section “Correlation network between mutations”. In the sentence: “If the mutation is the last one to be hovered before a variant’s development“, by “hovered” do you mean “appear” or a similar word?

Fig. 2. What does the direction of connecting arrows indicate?

Fig. 3. Please, add labels to the Y-axes of at least panels A and C. Although described in the text, the axes should have labels to facilitate interpretation. Also, the Y-axes need more ticks to rapidly determine the value of the corresponding ratios (just like in Fig. 4).

Page 8, lines 175-176. The text indicates that “a dot in the column is the ratio of specimens that contain each mutation.” Do you mean the ratio of specimens that contain all mutations in that column (it’s only one dot per column)? While this may seem obvious to the authors, it is not clear for the reader. Please clarify your description of these dots.

Fig 4. The legend says that axes are similar to those in Fig. 3, but Fig. 3 contains negative values in the X-axes and Fig. 4 has only positive numbers in the X-axes. The “similarity” is not obvious. There is no need to make the reader guess. Please explain your X-axes for this figure clearly. I recommend adding Y-axis labels to panels A, C and E. Although the interpretation of the X-axes becomes clearer later in the text, this should be clear directly in the figure legend.

Fig. 5. It is more accurate to say that “Both axes should be interpreted as in Fig. 4.”

Page 15, lines 319-322. The authors fail to make their point in the 4-line description about multiple sequence alignments and phylogenetic tree reconstruction.

Page 15, lines 322-323. I think the authors mean that the temporal order is statistically “inferred” or "deduced".

Unveiling the history of the variants is a huge task and important contribution. But I’m curious. I noticed that the authors made no effort to discuss the functional implications of the mutations as they appear in temporal order. Can the authors discuss how pre-variant mutations affect the function of the S protein? Are they essentially functionally neutral? Do they introduce important structural/mechanistic changes that explain the increase or decrease of the variants’ infectivity rates? This information may not be available, which is ok, but a connection with the literature discussing functional implications would emphasize the contribution of the authors.

Comments on the Quality of English Language

In general, the paper is understandable. However, I noticed several typos and grammatical errors. These are a few examples:

Page 11, line 217. Typo: “shareing”

Page 11, line 224. Typo: “tas”

Page 13, line 286. Grammar: “in the both datasets”

Page 15, line 350. Typo: “Columbia”

I recommend to read carefully the whole manuscript and correct all typographical/grammatical errors.

Round 2

Reviewer 2 Report

Comments and Suggestions for Authors

About network analyses in the present manuscript, the authors would like to put an
answer as follows. In Fig. 1, nodes are connected to others based on the numbers of
shared mutations. Fig. 1A shows that some pairs of variants have different mutations; e.g.,
Alpha and Delta share only a single mutation (D614G), while they have 9 and 7
mutations, respectively. In Fig. 2, the recall indices were calculated between mutations
based on the numbers of the specimens in which pairs of mutations are harbored. The
figure indicates that many mutations are shared with multiple variants. For these
purposes, quantitative indices for these networks were not applied.

I expected that the use of metrics like vertex degree, centrality and the like could ease the description of the node properties and relations (mutations or variants)

2. Thank you for the comment. As mentioned in the first and second sentences, the
usefulness of our study is to deduce the temporal order of the mutations and the
possibility of recombination between variants or subvariants. These points are based on
Figs 4 and 5.

3. I agree to the reviewer’s comment. Although the authors retrieved articles on prevariant
analyses of the virus’s mutation, it was difficult to find such articles. Therefore, the
authors referred to articles on the virus’s mutation analyses.

An easy search on PubMed or Google Scholar would provide many references on each variant from which to pick the most relevant.

4. Figs 1, 2, 4, and 5 are the snapshots from the Vcorn SARS-CoV-2 website constructed by
the authors. Therefore, their qualities may be limited on the manuscript for reviewing.

I understand. However, network data could be input into specialized software able to reproduce the network at a higher graphic quality.

5. Thank you for the useful comment. The many parts in Materials and Methods are based
on the authors’ previous reports (Ogata and Kitayama 2022). According to the comment,
the authors have added “The detailed procedure is described in our previous reports [9]”
in 2.4.

Yes, I understand. However, please, let me explain my doubts. I am aware that I may have completely misunderstood the sentence "The recall index of mutation A to mutation B
(equivalent to the precision index of mutation B to mutation A) was calculated as a ratio
of the number of specimens with both mutations A and B to the number of specimens
with only mutation A. If the index is one, all specimens with mutation A contain mutation B." (lines  94-97).

Now, let's suppose that we have 50 specimen with A and B and 50 with only A. The ratio is obviously 1. Why should I conclude that A is always associated with B? Maybe the definition of the recall index should be better clarified.

Reviewer 4 Report

Comments and Suggestions for Authors

1.    Page 1-2, lines 45-47. The sentence added by the authors to the introduction has grammatical errors (“there is no reports, …”).

2.    Page 5, Methods section “Data Processing”. The authors misunderstood my point about genome coverage. I apologize if I was not clear. Genome coverage refers to how many times each region of the genome was sequenced before producing the final version of the genome assembly. This information is available in the corresponding entries at NCBI. The higher the coverage (e.g., >50X), the more reliable the sequence to identify SNPs. If a genome was covered only 1X, then the probability that the differences observed with a reference genome are due to sequencing errors (as opposed to true mutations) becomes more significant. That is why deep sequencing is so popular for the study of variants. Did the authors filter the genomes by any measure of quality (there are several)? If all the genomes are products of deep sequencing techniques, then everything is ok. The authors need to show that they used reliable sequence data for their analysis. I say this because there are “complete” genomes with poor-quality sequences in public repositories. I found no information about this on their website or their BMC Genomic Data (2022) paper.

In the Vcorn website, the authors provide very short descriptions of their scripts. This makes it difficult to understand important technical details. For example, for “Pre-blast treatment” they only say: “chknat.pl & “mkpidsummary.pl create files that contains meta data of the SARS-CoV-2 samples.” But there is no description of the general procedures followed for data pre-processing. Were the genomes pre-filtered for qualityl? Did the authors simply work with all genomes regardless of their sequence quality?

This is nice work and I have no doubt the authors did everything correctly, but without an adequate description of technical aspects of data pre- and post-processing, readers may have a difficult time trusting and reproducing the results.

3.    Fig. 2 legend. The sentence added to explain the direction of the arrows is not using correct English. Please fix it.

4.    Page 16, lines 388-399. The sentence added by the authors is confusing. They say that mutations Y144S and Y145N are “less reported” and they provide two references [12,13]. What do these two reports say or conclude about those mutations? Do they suggest or describe the functional/mechanistic impact of the mutations or just their identification? For example, if only the identification of the mutations is discussed in those references, the authors can say that no reports have provided significant insights into the functional consequences of pre-variant mutations. A sentence or two about this would be enough.

5.    English. I was surprised to see that all the typos and grammatical errors from the original version of the manuscript are still there. The authors need to carefully check their manuscript for English to improve its readability. I recommend asking a native English speaker to proof-read your manuscript. These are some of the errors I can recall, but there are more:

Page 11, line 226 (originally 217). Typo: “shareing”

Page 11, line 233 (originally 224). Typo: “tas”

Page 13, line 295 (originally 286). Grammar: “in the both datasets”

Page 15, line 362 (originally 350). Typo: “Columbia”

Comments on the Quality of English Language

Unfortunately, the authors did not correct the typos and grammatical errors found in their original version of the manuscript. I added some of the examples of identified in my comments to the authors to make sure they see them.
